# Environmentally Relevant Levels of Antiepileptic Carbamazepine Altered Intestinal Microbial Composition and Metabolites in Amphibian Larvae

**DOI:** 10.3390/ijms25136950

**Published:** 2024-06-25

**Authors:** Wei Dang, Jin-Hui Zhang, Zi-Chun Cao, Jia-Meng Yang, Hong-Liang Lu

**Affiliations:** Key Laboratory of Hangzhou City for Ecosystem Protection and Restoration, School of Life and Environmental Sciences, Hangzhou Normal University, Hangzhou 311121, China; dangwei@hznu.edu.cn (W.D.); 2023111010092@stu.hznu.edu.cn (J.-H.Z.); 2022210314032@stu.hznu.edu.cn (Z.-C.C.); 2022111010080@stu.hznu.edu.cn (J.-M.Y.)

**Keywords:** *Pelophylax nigromaculatus* tadpole, antiepileptic pharmaceutical, microbiomic change, metabolomic response, energy metabolism, immunomodulatory property

## Abstract

There is growing concern about the potential ecological risks posed by pharmaceutical residues in the aquatic environment. However, our understanding of the toxic effects of antiepileptic pharmaceuticals, such as carbamazepine (CBZ), on aquatic animal larvae is still limited. In this study, the tadpoles of the black-spotted pond frog (*Pelophylax nigromaculatus*) were exposed to environmentally relevant concentrations of CBZ (0.3 and 3.0 μg/L) for 30 days, and their growth, intestinal microbial composition, and metabolites were investigated to assess the potential toxic effects of CBZ in non-targeted aquatic organisms. Some tadpoles died during exposure, but there was no significant among-group difference in the survival and growth rates. CBZ exposure significantly altered the composition of tadpole intestinal microbiota. Relative abundances of some bacterial genera (e.g., *Blautia*, *Prevotella*, *Bacillus*, *Microbacterium*, etc.) decreased, while others (e.g., *Paucibacter*, etc.) increased in CBZ-exposed tadpoles. Interestingly, CBZ-induced alterations in some bacteria might not necessarily lead to adverse outcomes for animals. Meanwhile, small molecular intestinal metabolites related to energy metabolism, and antioxidant and anti-inflammatory activities were also altered after exposure. Taken together, environmentally relevant levels of CBZ might alter the metabolic and immune performances of amphibian larvae by modifying the abundance of some specific bacteria and the level of metabolites in their intestines, thereby potentially causing a long-term effect on their fitness.

## 1. Introduction

Over the past few decades, large amounts of unmetabolized pharmaceuticals have been released into various natural environments, and their residual presence has posed a threat to the health and survival of non-targeted organisms [1,2]. Carbamazepine (CBZ) is a commonly prescribed antiepileptic pharmaceutical that can lower neural activity by blocking the sodium channels of excitatory neurons [1]. CBZ can persist in aquatic environments due to its slow degradation (a half-life of approximately 82 days [3]) and low removal efficiency from wastewater [4]), and its detectable concentration varies from 0.4 ng/L to 5 μg/L in surface water (the maximum concentration reported for China’s rivers is 1.09 μg/L) [5,6,7,8].

As a neuro-psychiatric pharmaceutical, it is predicable that CBZ exposure potentially affects the neurophysiology and behavior of non-targeted aquatic organisms. It has been reported that CBZ exposure alters the locomotor activity in aquatic arthropods (*Gammarus pulex* exposed to 0.01 μg/L) and planarians (*Schmidtea mediterranea* exposed to 1.0 μg/L), siphoning behavior of clams, and feeding behavior of fish (*Oryzias latipes* exposed to 6.15 mg/L) [9,10,11,12]. Additionally, CBZ can alter antioxidant ability as well as induce DNA damage and metabolic disorder in various aquatic organisms (e.g., aquatic insects, molluscs, crustaceans, fish, and others) [6,9,13,14,15,16,17,18,19]. For example, short-term CBZ exposure (up to 100 μg/L) could result in an increase in the bioconcentration and inhibition of the activities of some enzymes related to antioxidant and nervous system function in crustacean species *Daphnia magna* [6], while long-term exposure would alter the expression of enzymes/genes related to lipid metabolism and cause dyslipidemia and lipid droplet accumulation in fish species *Gobiocypris rarus* [18]. CBZ also exhibits homologous estrogenic activity, and it has been listed as an endocrine-disrupting chemical [20].

Multiple phenotypical, physiological, and behavioral endpoints have been involved in the toxicological risk assessment of the effects of CBZ on aquatic organisms, but omics analyses have rarely been applied to explore its toxic effects. Limited studies based on transcriptomic, proteomic, or metabolomic analysis have revealed extensive metabolic disturbances caused by CBZ exposure [21,22,23]. Recently, microbiomic analysis showed that CBZ exposure alters the intestinal microbial community composition and potentially affect the immune performance in zebrafish [8]. Generally, amphibian species are sensitive to environmental changes, including increasing pollutant levels, and their current population decline may partially have resulted from toxic exposure to environmental pollutants [24]. It is necessary to evaluate the toxicological effects of various emerging pollutants (e.g., CBZ) on amphibians to improve our understanding of their potential role in amphibian population decline. In this study, tadpoles of the black-spotted pond frog, *Pelophylax nigromaculatus*, were exposed to environmentally relevant concentrations of CBZ (0.3 and 3.0 μg/L) for 30 days, and the alterations in the intestinal microbial composition and metabolites were measured. Only limited information on CBZ-induced alterations in the intestinal microbial composition or metabolites from aquatic species is currently available for comparison [8]. The potential toxic effects of antiepileptic pharmaceutical were evaluated based on intestinal microbiomic and metabolomic analysis for the first time in amphibian larvae here. Following CBZ exposure, alterations in the microbial composition and the associated molecular metabolites were expected, which probably produce far-reaching consequences in non-targeted organisms. Our results contribute to promoting the understanding of the potential toxic risks of antiepileptic pharmaceuticals in aquatic environments.

## 2. Results

### 2.1. Tadpole Body Size and Developmental Stage

Some tadpoles (four and three individuals in the 0.3 µg/L- and 3.0 µg/L-exposed group, respectively) died during the exposure, but the mortality rate did not differ among groups (*χ*^2^ = 4.05, *df* = 2, *p* = 0.132). After a 30-day exposure, the body size (mass, CTRL: 0.342 ± 0.027 g; 0.3 µg/L: 0.340 ± 0.033 g; 3.0 µg/L: 0.323 ± 0.020 g) and developmental stage (CTRL: 30.8 ± 0.6; 0.3 µg/L: 31.9 ± 0.6; 3.0 µg/L: 31.6 ± 0.5) of the tadpoles did not differ significantly among groups (body mass, *F*_2, 8.6_ = 0.01, *p* = 0.989; developmental stage, *F*_2, 9.1_ = 0.94, *p* = 0.429).

### 2.2. Tadpole Intestinal Microbiota

The obtained 16S rDNA gene sequences generated 2024 OTUs (Appendix A). No significant differences were found in the alpha diversity index of the OTU level for the tadpole intestinal microbiota among groups (Chao index, CTRL: 840.6 ± 210.1; 0.3 µg/L: 833.2 ± 96.5; 3.0 µg/L: 646.2 ± 92.2, *H* = 1.34, *p* = 0.512; Shannon index, CTRL: 3.54 ± 0.48; 0.3 µg/L: 3.20 ± 0.17; 3.0 µg/L: 3.35 ± 0.12, *H* = 0.26, *p* = 0.878).

CBZ exposure altered the intestinal microbial composition of the tadpoles (Appendix A). The relative abundances of some bacterial genera [e.g., *Blautia* (belonging to phylum Firmicutes, family Lachnospiraceae), *Bacillus* (Bacillaceae), *Rhodococcus* (Actinobacteria, Actinomycetaceae), *Fusobacterium* (Fusobacteria, Fusobacteriaceae) etc.] decreased significantly, while those of some genera [e.g., *Paucibacter* (Proteobacteria, Comamonadaceae), *Crenobacter* (Neisseriaceae), *Kinneretia* (Comamonadaceae), *Pirellula* (Planctomycetes, Planctomycetaceae)] increased in the CBZ-exposed tadpoles (Figure 1). Some bacterial genera were primarily found in the CTRL group [e.g., *Prevotella* (Bacteroidetes, Prevotellaceae), *Stomatobaculum* (Firmicutes, Lachnospiraceae), *Phyllobacterium* (Proteobacteria, Phyllobacteriaceae)] or only in CBZ-exposed tadpoles [e.g., *Alsobacter* (Proteobacteria, Alsobacteraceae)], which accounted for a small proportion (<0.01%, Figure 1). The functional pathways of intestinal microbiota might be altered by CBZ exposure. At the second level, the abundances of the predicted KEGG pathways, including amino acid and lipid metabolisms, as well as membrane transport decreased, while those of the pathways associated with the immune system, environmental adaptation, folding, sorting and degradation, and neurodegenerative disease increased in the CBZ-exposed tadpoles (Appendix A). At the third level, the abundances of some pathways (e.g., valine, leucine, and isoleucine biosynthesis; histidine metabolism; arginine and proline metabolism; etc.) decreased, while those of others (e.g., biotin metabolism, lipopolysaccharide biosynthesis, sulfur relay system, etc.) increased in the CBZ-exposed tadpoles (Figure 2).

### 2.3. Tadpole Intestinal Metabolite

The multivariate analyses [principal component analysis (PCA) and partial least squares discriminant analysis (PLS-DA)] of the intestinal metabonomic data identified obvious separations among the groups (Appendix A). The univariate analyses for identified intestinal metabolites showed that some metabolites (e.g., leucine, creatine, cortisol, succinic acid, indole, etc.) decreased, while others (e.g., phenylalanine, stearolic acid, phenylpyruvic acid, etc.) increased in the CBZ-exposed tadpoles (Table 1, Figure 3).

The correlation analysis of significantly changed bacterial genera and metabolites showed that some bacteria were correlated with multiple metabolites, while some were solely correlated with an identified metabolite. For example, the abundances of bacterial genera *Rhodococcus*, *Aliihoeflea,* and *Phyllobacterium* were positively correlated with the levels of indole, cortisol, succinic acid, etc., but negatively with the level of phenylalanine. The abundance of *Blautia* (or *Prevotella*, *Fusobacterium*, etc.) was solely correlated with the level of uridine (Figure 4).

## 3. Discussion

Physiologically, the intestinal microbial and metabolic responses to antiepileptic pharmaceutical CBZ exposure at environmentally relevant concentrations were investigated in *Pelophylax nigromaculatus* tadpoles in this study. Behavioral, physiological, and biochemical disorders induced by CBZ exposure have been found in some aquatic organisms, including polychaetes, molluscs, crustaceans, and fish [6,9,13,16,17,18,22,25,26,27]. Some studies have also evaluated the potential toxicity of CBZ in the larvae of amphibian species, including *Xenopus laevis*, *Limnodynastes peronii*, *Rana dalmatina*, and *Bufo bufo* [15,28,29], but none of them considered more sensitive toxicological endpoints, such as through biochemical and molecular testing. This study integrated intestinal microbiomic and metabonomic alterations for the first time to investigate the toxic effects of CBZ in an amphibian species. No significant differences in the mortality rate, body mass, or developmental stage of the tadpoles were found in this study, indicating a limited impact of CBZ on the survival and growth of amphibian larvae. Actually, even after exposure to higher concentrations of CBZ (100 μg/L in *L. peronii*, 50 μg/L in *R. dalmatina*), survival and growth differences could not be observed in other amphibian species [15,28]. No significantly increased mortality or altered growth rate was observed in some aquatic invertebrates exposed to low levels of CBZ (≤9 μg/L) [9,25]. CBZ-induced growth inhibition in aquatic invertebrates might occur under exposure to unrealistically high concentrations (>9 mg/L) [1].

Intestinal microbiomic and metabonomic alterations potentially revealed some impacts of CBZ exposure on *P. nigromaculatus* larvae. Although the intestinal microbial α diversity did not differ among groups, the composition of the intestinal microbiota could have changed significantly in the CBZ-exposed tadpoles. For example, the abundances of some bacterial genera, which have anti-inflammatory and immunomodulatory properties (e.g., *Blautia*, *Prevotella*, *Bacillus*) and participate in nutrient uptake and production (e.g., *Microbacterium* polysaccharide production, *Cetobacterium* improvement in glucose homeostasis, vitamin synthesis), significantly decreased in the CBZ-exposed tadpoles. These alterations in specific bacteria after exposure might affect animal body state [30,31]. Unexpectedly, the abundances of some pathogenic bacteria, such as *Fusobacterium* and *Granulicatella*, were found to decrease (while that of *Kinneretia*, which has potential antimicrobial properties, increased) in the CBZ-exposed tadpoles. The alterations in the abundances of these bacterial genera seemed to be beneficial for animal health, and their subsequent outcome should be investigated in future studies. Contrarily, decreased abundances of beneficial bacterial genera (e.g., *Cetobacterium*) but no significant change in intestinal microbial diversity after feeding with CBZ diet were observed in zebrafish [8]. The discrepant results across studies might be due to the differences in treatment method. Aqueous exposure to low levels of CBZ might not necessarily lead to intestinal microbial imbalance in tadpoles. The increased abundance of some bacteria that potentially contributed to CBZ degradation was observed in CBZ-exposed soil insects [32], but such a change was not evident in this study.

Intestinal microbial functional metabolism can be modified by microbial composition alterations [30]. PICRUSt functional prediction analysis revealed the disruption of multiple metabolic pathways in the CBZ-exposed tadpoles. The decreased abundance of the pathway of valine, leucine, and isoleucine biosynthesis suggested lower intestinal levels of branched-chain amino acids (BCAAs; the decreased intestinal leucine level was confirmed by the metabolite profiles) in the CBZ-exposed tadpoles. Deficiency in BCAAs in animals might influence their development and growth, and lead to a high incidence of external deformity [30,33]. CBZ contains an aromatic hydrocarbon structure. The abundance of those intestinal bacteria that contribute to degrading polycyclic (and/or heterocyclic) aromatic hydrocarbons, as well as the related degradation pathways, should increase after CBZ exposure [32]. Contrarily, the predicted abundance of the pathway of polycyclic aromatic hydrocarbon degradation was shown to decrease. Similarly, the study of Wang et al. [32] also found CBZ exposure enhanced antibiotic (such as beta-lactam and tetracycline) resistance. However, inconsistent outcomes from the predicted abundance of beta-lactam resistance and the tetracycline biosynthesis pathway were observed here.

Small molecular metabolites produced by intestinal bacteria affect animal health [34]. The results of intestinal metabolite alterations further indicated the potential metabolic disruptions caused by CBZ exposure in animals. A reduced succinate level might affect the functioning of the tricarboxylic acid (TCA) cycle, and disturb energy metabolism. Some amino acids (such as leucine, phenylalanine) can be converted into intermediates (citrate, fumarate, etc.) of the TCA cycle [35,36]. Alterations in these amino acids would bring about an indirect impact on energy metabolism, although an opposite trend in the changes in the leucine and phenylalanine levels was shown here. A disturbance of energy metabolism caused by CBZ exposure was also reported in crustacean (*Daphnia magna*, through amino acid alterations) and fish species (*Gobiocypris rarus*) [22,37]. The level of cortisol, a stress hormone, could be expected to reduce under antiepileptic CBZ treatment [38]. This seems to be consistent with the changing trend of the intestinal cortisol level in the CBZ-exposed tadpoles. Conversely, no significant alteration in whole-body cortisol levels was shown in fish (*Jenynsia multidentata*) exposed to 200 μg/L of CBZ [39]. Taurine, indole, and their derivatives have been shown to have antioxidant and anti-inflammatory effects [40,41]. Decreased intestinal taurine and indole levels in CBZ-exposed tadpoles might result in a weakened resistance to pathogenic bacteria, although no significantly increased abundances of pathogenic bacteria were observed here. Phenylpyruvic acid can be produced by intestinal bacteria (such as *Bacteroides*, *Proteus*, despite the lack of change in bacterial abundance) from phenylalanine with the catalysis of phenylalanine ammonia lyase. Increased phenylpyruvic acid levels would be harmful to health and potentially increase inflammatory responses [42].

## 4. Materials and Methods

Carbamazepine (CBZ, >99% purity, CAS no. 298-46-4) and other chemical reagents were purchased from the Sinopharm Chemical Reagent Co., Ltd. (Shanghai, China). CBZ was dissolved in dimethyl sulfoxide (DMSO) and diluted in distilled water to prepare stock solutions.

In April of 2023, *P. nigromaculatus* tadpoles (at Gosner developmental stage 20–21) from Jingzhou (Hubei Province, China) were collected and kept in aquaria that were placed in a room at a temperature of 25 ± 1 °C and photoperiod of 12 h light:12 h dark. They were fed with powdered food. After being acclimated to the laboratory conditions for 2 weeks, 84 tadpoles (approximately at Gosner stage 26) were randomly selected and allocated equally to three experimental treatments. Each treatment group included 4 replicate tanks with 7 tadpoles per tank in a static system. Each tank contained 3 L of dechlorinated tap water, which was gently oxygenated with a mini aquarium pump. The CBZ concentrations of 0.3 and 3.0 μg/L in this study were selected with reference to previous studies in molluscs and fish [14,25,43]. Into the 4 tanks of the control (referred to as CTRL) group, an equal volume of DMSO was added. Water quality parameters were temperature, 26.2 ± 0.1 °C; pH, 7.51 ± 0.08; electrical conductivity, 142.2 ± 1.2 μs/cm; dissolved oxygen, 6.41 ± 0.08 mg/L; NH_3_-N, 0.10 ± 0.03 mg/L. The general care of tadpoles and the exposure process were described in previous studies [30,44].

During exposure, dead tadpoles were removed from the tanks. After 30-day exposure, tadpoles were euthanized and weighed for mass, and we determined the developmental stage; of these, 15 tadpoles in each treatment (from different tanks) were dissected for intestinal contents. Intestinal content samples from 3 tadpoles within the same treatment group were mixed and used for each of the microbiomic and metabolomic analyses. The animal experimental procedures were approved by the Animal Care and Ethics Committee of Hangzhou Normal University (2023051). The Pearson’s chi-squared (*χ*^2^) test was performed to test the difference in tadpole mortality rate, and mixed-model analysis of variance (ANOVA, with treatment as the fixed factor and tank identity as the random factor) was performed to test the differences in body size (mass) and developmental stage.

Intestinal microbiomic and metabolomic analyses were completed by Hangzhou Kaitai Biotechnology Co., Ltd. (Hangzhou, China), and their detailed procedures were described previously [30,43]. Briefly, total bacterial DNA was extracted from a portion of an intestinal content sample, and the V3-V4 variable region of 16S rDNA gene was amplified using primers (B341F: 5′-CCTACGGGNGGCWGCAG-3′, and B785R: 5′-GACTACHVGGGTATCTAATCC-3′). Amplified product was purified, qualified, and finally sequenced on the Illumina MiSeq^®^ PE300 platform (Illumina Inc., San Diego, CA, USA). After being quality-filtered and denoised, obtained bacterial sequences were clustered into operational taxonomic units (OTUs), which were annotated based on the Silva database, and alpha diversity (Chao and Shannon) indices of OUT level were calculated. The function of intestinal microbiota was predicted using PICRUSt2 based on the KEGG database. Non-parametric Kruskal–Wallis test was performed to test among-group differences in alpha diversity indices of intestinal microbiota, as well as relative abundances of identified intestinal bacteria and KEGG pathways.

Another portion of intestinal content sample (approximately 100 mg) was used for metabolomic analysis. Briefly, extract solution was added to the samples, which were ground and incubated. After centrifugation, their supernatants were collected and blow-dried, and finally products were redissolved and filtrated. Liquid chromatography–mass spectrometry (LC–MS)-based metabolomic profiling was performed on a Vanquish UPLC system coupled to an Orbitrap Exploris 120 mass spectrometer (Thermo Fisher Scientific Inc., Waltham, MA, USA). Raw LC-MS data were converted using Proteowizard v3.0.8789, processed using the XCMS package in R version 4.4, and normalized and further analyzed using the MetaboAnalyst web-based platform. Both PCA and PLS-DA were performed to analyze among-group variations in metabolomic profiles. One-way ANOVA was performed to test among-group differences in spectral areas for some identified metabolites. Further to this, correlation analysis was performed to test the correlation between some significantly different intestinal bacterial genera and metabolites. Before parametric analyses, data were tested for normality using the Kolmogorov–Smirnov test and homogeneity of variances using the Levene’s test.

## 5. Conclusions

Limited impacts of antiepileptic pharmaceutical CBZ exposure at environmentally relevant concentrations on the survival and growth of amphibian larvae were indicated by the lack of observable phenotypic alterations in the exposed tadpoles. Despite decreased abundances of some beneficial bacteria, our observed alterations in intestinal bacterial abundances suggested that exposure to low levels of CBZ might not necessarily have adverse effects on the microbial community of tadpoles. Obvious intestinal metabolite alterations could also be observed, potentially reflecting a CBZ-caused imbalance in metabolic homeostasis in this species. Further to this, decreased levels of some intestinal metabolites with anti-inflammatory and immunomodulatory effects, and related to energy metabolism, might affect the immune and metabolic performance of CBZ-exposed tadpoles, and thus reduce their survival and growth during their subsequent life stages. In order to reveal such potential consequences of CBZ exposure, the fitness-related performance of exposed tadpoles should be investigated over a longer period of time (even in the post-metamorphic stage) in future studies.

## Figures and Tables

**Figure 1 ijms-25-06950-f001:**
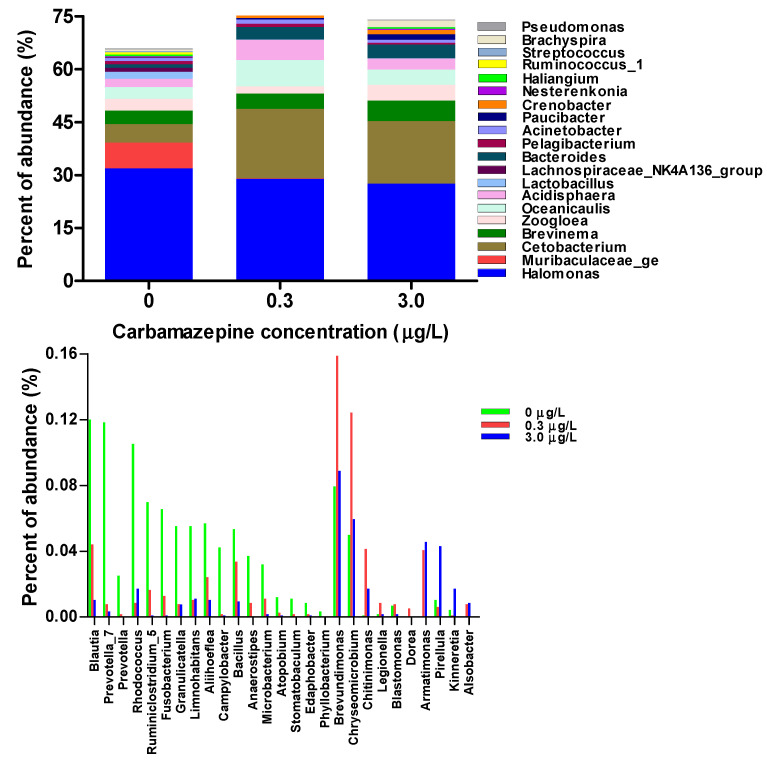
Relative abundances of primary bacterial genera and some significantly changed bacterial genera in *Pelophylax nigromaculatus* tadpoles exposed to 0 (CTRL), 0.3, and 3.0 μg/L of carbamazepine.

**Figure 2 ijms-25-06950-f002:**
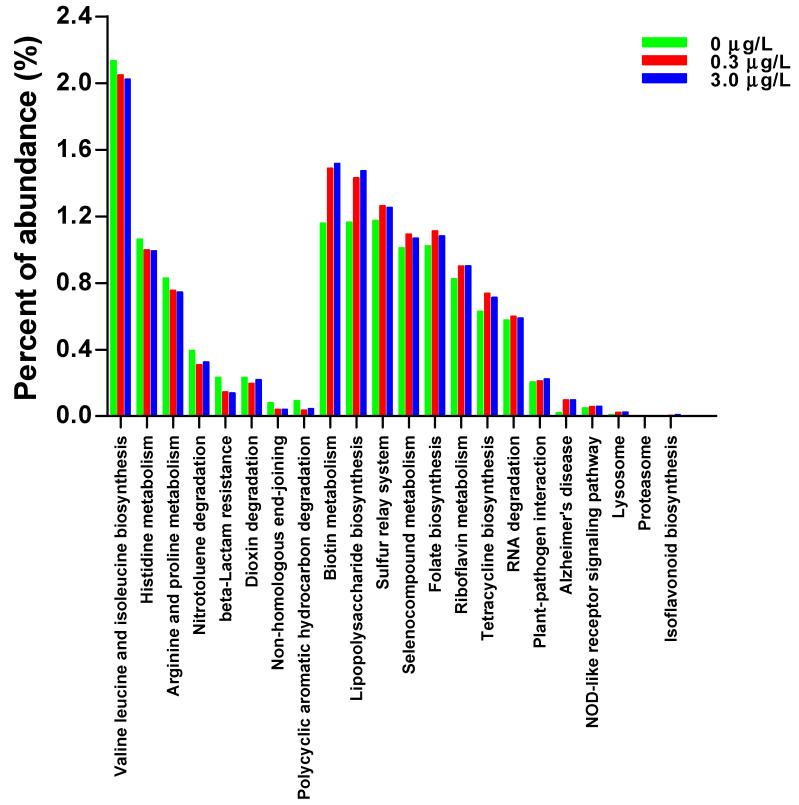
Relative abundances of predicted functional pathways with significant among-group differences in *Pelophylax nigromaculatus* tadpoles exposed to 0 (CTRL), 0.3, and 3.0 μg/L of carbamazepine.

**Figure 3 ijms-25-06950-f003:**
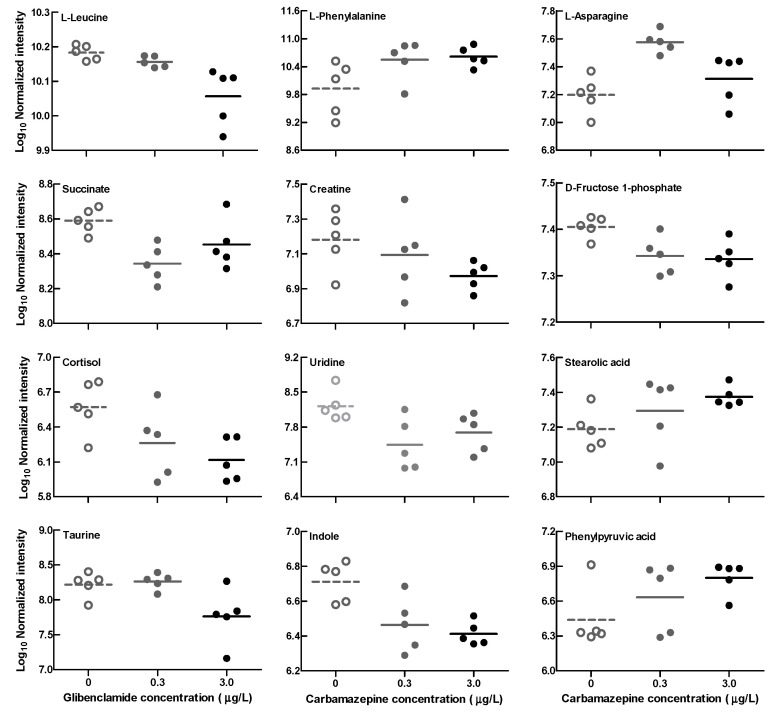
Spectral area for some significantly changed intestinal metabolites in *Pelophylax nigromaculatus* tadpoles exposed to 0 (CTRL), 0.3, and 3.0 μg/L of carbamazepine. Each dot represented data from a single sample.

**Figure 4 ijms-25-06950-f004:**
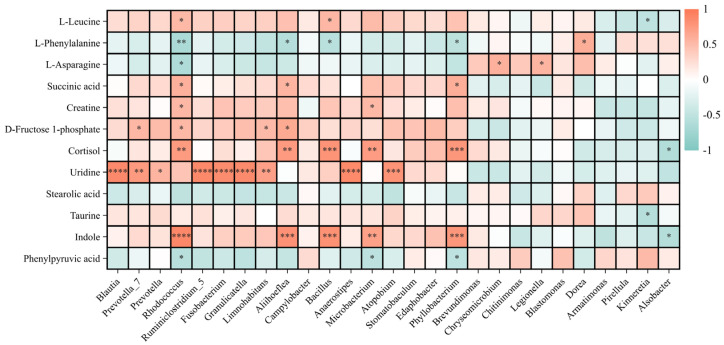
The correlation between significantly changed bacterial genera and significantly changed metabolites in *Pelophylax nigromaculatus* tadpoles exposed to 0 (CTRL), 0.3, and 3.0 μg/L of carbamazepine (* *p* < 0.05, ** *p* < 0.01, *** *p* < 0.001, **** *p* < 0.0001).

**Table 1 ijms-25-06950-t001:** Several identified metabolites in the intestine of *Pelophylax nigromaculatus* tadpoles exposed to 0, 0.3, and 3.0 μg/L of carbamazepine, and their fold changes (FCs) and associated *p*-values (* *p* < 0.05, ** *p* < 0.01).

	0.3 μg/L vs. CTRL		3.0 μg/L vs. CTRL	
Metabolite	Log_2_(FC)	*p*	Log_2_(FC)	*p*
L-Leucine	−0.09		−0.40	**
L-Phenylalanine	1.67	*	1.62	*
L-Asparagine	1.22	**	0.42	
Succinate	−0.80	**	−0.40	
Creatine	−0.22		−0.75	*
D-Fructose 1-phosphate	−0.20	*	−0.23	*
Cortisol	−0.89		−1.55	*
Uridine	−2.11	*	−1.72	*
Stearolic acid	0.42		0.59	*
Taurine	0.09		−1.18	*
Indole	−0.79	*	−1.02	**
Phenylpyruvic acid	0.61		0.98	*

## Data Availability

All data generated by this study are available in this manuscript and the accompanying Appendix A.

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
