# Peer review of "Environmentally Relevant Levels of Antiepileptic Carbamazepine Altered Intestinal Microbial Composition and Metabolites in Amphibian Larvae"

_ijms, 2024, doi:10.3390/ijms25136950_

Round 1

Reviewer 1 Report

Comments and Suggestions for Authors

The work presented in this manuscript investigated effects on microflora and metabolite levels due to exposure to two levels of CBZ. The authors found no apparent adverse effects but compositional variation of microflora as well as metabolite levels.

The manuscript is well organized and well written.

Although the relevance of the experimental result of this work is limited for the assessment of effect of CBZ on aquatic environmental, accumulation of this kind of descriptive information might be useful in the future ecological risk assessment.

The most serious criticism to this manuscript is that the authors’ standpoint of inference of result was that the effects found in this experiment was toxic effects. There were no apparent toxic symptoms in tadpoles by the exposure except for small variation in microflora composition. There is no reason to believe that the small variation found in the exposed tadpoles was adverse. Indeed, the authors state decreased abundance of pathogenetic bacteria in exposed tadpoles, but still they suggest as if the microflora change was adverse throughout the Discussion section. The authors’ standpoint seems to be too biased. They should interpret the results from neutral standpoint. The Discussion and Conclusion sections should be re-written.

The followings are some minor points for consideration.

1.      Describe clearly if tadpoles were exposed to CBZ in flow condition or static condition.

2.      The concentration of CBZ in the river where the tadpoles for experiment were caught should be reported.

Author Response

The work presented in this manuscript investigated effects on microflora and metabolite levels due to exposure to two levels of CBZ. The authors found no apparent adverse effects but compositional variation of microflora as well as metabolite levels.

The manuscript is well organized and well written.

Although the relevance of the experimental result of this work is limited for the assessment of effect of CBZ on aquatic environmental, accumulation of this kind of descriptive information might be useful in the future ecological risk assessment.

(Thanks for the comments.)

The most serious criticism to this manuscript is that the authors’ standpoint of inference of result was that the effects found in this experiment was toxic effects. There were no apparent toxic symptoms in tadpoles by the exposure except for small variation in microflora composition. There is no reason to believe that the small variation found in the exposed tadpoles was adverse. Indeed, the authors state decreased abundance of pathogenetic bacteria in exposed tadpoles, but still they suggest as if the microflora change was adverse throughout the Discussion section. The authors’ standpoint seems to be too biased. They should interpret the results from neutral standpoint. The Discussion and Conclusion sections should be re-written.

(We are very grateful for this constructive comment. We have made extensive revisions in the sections of Discussion and Conclusion.)

  1. Describe clearly if tadpoles were exposed to CBZ in flow condition or static condition.

(Revised. Tadpoles were maintained in tanks in a static system, but water in tanks was aerated with a mini aquarium pump.)

  1. The concentration of CBZ in the river where the tadpoles for experiment were caught should be reported.

(We agree with the reviewer’s comment. Unfortunately. We didn't measure the concentration of CBZ in the waterbody, where tadpoles were collected. Generally, the concentration of CBZ in the natural waters is at nanogram per liter levels, Liu and Wong, 2013, Environ. Int. 59, 208–224; Xiang et al., 2021, Ecotoxicol. Environ. Saf. 213, 112044. We would consider measuring the levels of environmental pollutants in waterbodies where experimental animals were collected in future studies.)

Reviewer 2 Report

Comments and Suggestions for Authors

The article requires significant additions and improvements:

1. Write the abstract again, please: from a reference to the problem, through methodology, key results and emphasizing the novelty of your research.

2. In the introduction, add laconic references to literature (e.g. line 44). You provide various, cumulative references, so describe their scientific contributions in a full and interesting way.

3. In the last paragraph of the introduction, complete the aim of the work and emphasize the novelty.

4. The figures you included in your article are too small to be interpreted and may be published in this form. Separate them, please.

5. Complete the analysis of the obtained data, both in diagrams and in tables. They are too laconic. Similarly, the conclusions are too general. 6. Unfortunately, the article cannot be published in its current form. You need to make an effort to substantively complete the analysis of the results and improve the graphic design.

Comments on the Quality of English Language

Minor editing of English language required

Author Response

The article requires significant additions and improvements:

  1. Write the abstract again, please: from a reference to the problem, through methodology, key results and emphasizing the novelty of your research.

(Revised. The abstract has been written again according to reviewers’ kind suggestions.)

  1. In the introduction, add laconic references to literature (e.g. line 44). You provide various, cumulative references, so describe their scientific contributions in a full and interesting way.

(Revised. The specific research cited have been analyzed, summarized, and presented in the article.)

  1. In the last paragraph of the introduction, complete the aim of the work and emphasize the novelty.

(Thanks for comment. We have made adjustments at the end of the introduction section based on the friendly suggestion from the reviewers.)

  1. The figures you included in your article are too small to be interpreted and may be published in this form. Separate them, please.

(Revised. The figure had been split into two, and the resolution of the figures were adjusted to 300dpi, making it clearer.)

  1. Complete the analysis of the obtained data, both in diagrams and in tables. They are too laconic. Similarly, the conclusions are too general.

(Thanks for comment. We checked our data and statistical results. The conclusions were written again according to reviewers’ kind suggestions.)

Round 2

Reviewer 1 Report

Comments and Suggestions for Authors

The authors have revised the manuscript to a satisfactory extent. The discussion about lack of CBZ measurement of the water where the tadpoles were collected for this experiment was hopefully given: particularly, what influence would have been resulted if CBZ level was higher.